# How Sleep Affects Recovery and Performance in Basketball: A Systematic Review

**DOI:** 10.3390/brainsci12111570

**Published:** 2022-11-18

**Authors:** Javier Ochoa-Lácar, Meeta Singh, Stephen P. Bird, Jonathan Charest, Thomas Huyghe, Julio Calleja-González

**Affiliations:** 1Independent Researcher, 31011 Pamplona, Spain; 2Henry Ford Sleep Disorders Center, Detroit, MI 48202, USA; 3School of Health and Medical Sciences, University of Southern Queensland, Ipswich, QLD 4350, Australia; 4Centre for Sleep & Human Performance, Calgary, AB T2X 3V4, Canada; 5Department of Kinesiology, University of Calgary, Calgary, AB T2N 1N4, Canada; 6Department of Sports Science, Universidad Católica de Murcia, 30107 Murcia, Spain; 7Department of Physical Education and Sport, University of the Basque Country, 48940 Vitoria, Spain

**Keywords:** sports performance, basketball, sleep, recovery, travel, circadian rhythms

## Abstract

Background: Sleep is considered an essential component related to physiological and psychological recovery in athletes and particularly in basketball, given the impact of condensed travel and game schedules on player health and performance. Objective: The aim of this systematic review is to examine studies published to date on sleep and basketball performance. Methodology: All scientific articles that reported a relationship between sleep and its possible impact on performance in basketball are included. The research processes followed the PRISMA criteria, and the relevant articles were extracted (PubMed, WOS, Scopus) as of December 31, 2021. Results: Twenty-eight articles were selected for inclusion and data extraction, with 27 demonstrating that sleep is a vital component in the recovery of basketball players and their corresponding on-court performance. Three central themes that we identified: (1) the quality and extension of sleep (the better quality and more extension of sleep, better performance and lower probability of injury); (2) influence of the players circadian rhythm (travel and game scheduling do not currently facilitate or take this into account); and (3) higher training loads and/or increased stress may jeopardize a subsequent good night’s sleep, which should be taken into account when scheduling practices and workouts. Conclusion: The current systematic review regarding sleep and basketball performance and highlights that there is a strong relationship between both variables. Collectively, the evidence supports the critical influence of sleep on player recovery and basketball performance and risk for injury.

## 1. Introduction

Basketball is an intermittent sport made up of high intensity repeated accelerations, decelerations, change of direction and jump landings, which places a high demand on players, both physically and cognitively [1]. In this team sport, the main physical actions that determine performance are accelerations and decelerations, changes of direction, and vertical jumps [2]. In addition to the importance of developing explosive power [3,4,5], aerobic capacity plays a key role, acting as a basis for the recovery of the player [6] and optimal recovery with methods with scientific evidence (foam roll, cold water Immersion and others) after practices and matches play a key role on team sport performance [7].

The scientific literature concludes that sports performance is not limited to the training performed by the athlete, but other factors as well, including both extrinsic and intrinsic variables [8,9]. In this sense, individualized recovery strategies are considered instrumental in achieving and sustaining the highest possible level of performance [10,11], and specifically in elite basketball ecosystems [12], where the players spend a much greater proportion of their time recovering than they do in training [6]. For instance, recovery strategies (Foam roll, ergo nutritional aids, cold water Immersion and others) help basketball players decrease their feeling of fatigue, an attribute that is directly related to athletic performance [13], considering the balance between *happiness* vs. *Wellness*, during the recovery process in high performance Sport [14]. 

One recovery strategy in particular, sleep, is commonly described as a determining factor in the recovery of players after a competition or after intense training session [15,16]. As such, it is theorized that a lack of sleep will foster negative consequences for performance. This appears to be critical in sports where motor coordination, decision-making and aerobic capacity are all fundamentally important [16,17,18,19,20]. 

Indeed, basketball is a prime example of the abovementioned characteristics [21]. To date, literature examining sleep and recovery on basketball considers sleep as one of the main recovery strategies available to players, alongside nutrition and hydration [2,6]. These factors are also considered as the most efficient recovery practices by the players themselves [22].

Furthermore, lack of sufficient regeneration may lead to a loss of performance due to the accumulation of holistic stress in the athlete’s body [23]. There are also publications that relate lack of sleep with negative effects on performance, both physically and cognitively, and injury [24]. In general, science presents the perverse effects on sports performance due the lack of quality sleep and, consequently, there is a growing interest to improve the sleep habits of athletes [15,25].

As such, it is essential to understand why basketball ecosystems, specially at the highest level, are so disruptive for sleep and therefore, for recovery. Due to diverse factors, such as scheduling, traveling, late practices and games, and electronic devices use, the probabilities of a solid and constant sleep, night after night, are scarce for elite basketball players. Finally, novel scientific studies examining the impact of air traveling direction, flight time, flight duration, average flight altitude, frequency and magnitude of height changes during flight, air cabin conditions, oxygen saturation levels, and athlete chronotype are warranted to help paint a clearer picture on how different stressors impact wellness and performance due to traveling [26]. These factors may represent significant problems in terms of athlete health and performance.

Considering the large body of knowledge on sleep and sport performance, surprisingly, for the best of the authors’ knowledge, there remains limited research that systematically and critically examines the health and performance effects of daytime sleepiness in basketball. 

## 2. Objectives

The main purpose of this systematic review was to explore the underlying physiological and psychological effects of sleep on health and performance in basketball. Secondly, we aim to extend previous literature by providing up to date, best-practice recommendations for sleep during a competitive basketball season.

## 3. Methodology and Procedures 

### 3.1. Search Strategy

This systematic review focused on the relationship between sleep, health and basketball player performance and followed the guidelines established by Preferred Reporting Items for Systematic Review and Meta-Analyses [27,28,29]. The main parameter used for the inclusion-exclusion of articles has been the PICOS Model [30], as well as the OLE scale [31]. We have used the following inclusion criteria: P (Population): professional and amateur basketball players; I (Intervention): effect of sleep on performance; C (Control): does not apply; O (Outcome): consequence of sleep-in basketball performance; S (Study Design): all kinds.

Therefore, a structured computer search was undertaken in different specialized online database media (PubMed, Web of Science [database that includes other databases such as BCI, BIOSIS, CCC, DIIDW, INSPEC, KJD, RSCI and SCIELO, all of high quality], Scopus). This search ended on 31 December 2021. The search terms included a mix of medical subject headings (MeSH) and free-text words for key concepts related to recovery, young, basketball, and players. The following search equation was used to find the relevant articles: ["sleep basketball" (MeSH Terms) OR"sleep" (all fields)] AND ["basketball" (all fields)]. We have also done it in Spanish, using the formula ["sleep basketball " (MeSH TERMS) OR "sleep" (all the fields)] AND ["Basketball (all fields)] OR ["sleep" (all fields)] AND ["Basketball" (all fields)]. There were no filters applied to the athlete’s physical fitness level, race, or age to increase the power of the analysis. The search has been carried out independently by two authors, JOL and J.C-G, disagreements were discussed with the third author (S.B)

The search terms selected were limited to sleep and basketball. No other terms were used additionally to increase the power of analysis. Through this equation, all relevant articles from this given field were obtained. The reference sections of all identified articles were also examined using the "Snowball Methods" strategy, based on the examination of reference sections of the selected articles [32].

This search method allowed us to apply the PICOS Model [33] on all the results obtained, which was the main inclusion-exclusion criterion of the articles for analysis. These database reviews have been carried out with the data obtained up to December 2021. Regarding the data collection process, the methodology used was the reading of each of the selected articles by the authors of this review. After carefully reviewing all articles, the authors finally excluded 3 more articles (2 opinion articles and one narrative review [18,26,34]). Hence, 25 articles were included in this study. The authors presented a summary table with the most relevant information of each study, in an easy-to-digest format, which in turn, helped the authors analyze, categorize and classify all the data necessary to carry out this review (Table 3). Titles and abstracts were selected to review the full text. Two authors (J.O-L. and J.C-G.) searched for independently published studies, and disagreements on all outcomes were resolved by a third author (S.B) 

### 3.2. Inclusion and Exclusion Criteria

Studies included in this systematic review had to meet the following inclusion criteria: (I) the study population comprised basketball players (College level or above); (II) sleep measurements were recorded; (III) the effects of sleep were compared with one or more indicators of basketball performance; and (IV) study designs included all possible designs: quantitative, qualitative, and/or mixed-method model with experimental, quasi-experimental, and/or non-experimental research design, utilizing primary and/or secondary data sources. 

Again, in our case, the PICOS model would be as follows: Professional, semi-professional, college, amateur and disabled basketball players (P); Effects of sleep on performance (I); Not applicable (C); Relationship of sleep with specific performance in basketball (O); and all, including quantitative, qualitative, and/or mixed-method model witexperimental, quasi-experimental, and/or non-experimental research design, utilizing primary and/or secondary data sources (S). 

All search titles and abstracts were collated to identify duplicates and possible missing studies. Titles and abstracts were screened for further full-text review. The search for published studies was conducted independently by two authors (JOL and h JC-G) and disagreements were resolved through discussions with another author (M.S).

We have also applied the OLE scale [30], as follows (Table 1):

We have applied the OLE scale to the 25 studies analyzed, being such that (Table 2):

### 3.3. Selection of Studies

Once the inclusion and exclusion criterion has been applied, the information obtained was evaluated by the two authors (JOL and JC-G) to define their selection. The year of publication, the authors, the sample size, the type of study and the characteristics of each of them were taken into account. Afterwards, all the relevant information was put into an Excel sheet (Microsoft Inc, Seattle, WA, USA) independently by two authors (JOL and JC-G) and the disagreements were resolved through discussions with the third author (J.CH). 

### 3.4. Extraction of Information

Once the works to be analyzed were selected, the following information was designated from each of them: year, author and publication; sample size; applied methodology; variables analyzed; results.

In reference to the methodology applied for each of the studies analyzed, 1 of these 25 articles were a systematic review of the literature [35], and there were also 2 articles using questionnaires for the participants [22,36]. Apart from these, the rest of the articles studied were based on mixed linear models of statistical analysis, where we highlight the use of Bonferroni (2 articles: [37,38]), MANOVA (4 articles: [39,40,41,42]), Student’s T (4 articles: [38,43,44,45]), regression analysis (7 articles: [46,47,48,49,50,51,52] and the use of Spearman correlation [45].

### 3.5. Assessment of the Quality of the Studies

Considering the potential limitations of the studies included in this systematic review, and in order to draw reliable conclusions, the Cochrane Collaboration Guidelines were followed [27]. Thus, the two authors independently assessed methodological quality and risk of bias (JOL and JC-G), while disagreements were resolved by a third-party assessment (T.H). 

In the Cochrane Risk of Bias tool [28], the following elements were included and divided into different domains: (1) selection bias (elements, random sequencing, allocation and concealment), (2) performance bias (blinding of participants and staff), (3) detection bias (blinding of outcome assessment), (4) assertion bias (incomplete outcome data), (5) reporting bias (selective reporting) and (6) other bias (other sources of bias). The assessment of risk of bias was characterized as (a) low-risk (plausible bias that is not likely to seriously alter the results), (b) unclear risk (plausible bias that raises some doubt about the results), or (c) high risk (plausible bias that seriously weakens confidence in the results).

Finally, observational studies such as those reviewed in this paper may suffer from this possibility; In addition, there is the possible limitation of the existence of different ways of reacting to sleep in each individual (since it is impossible to know whether the athletes with the best sleep outcomes are performing better on the court or they just happen to be the best players.) However, the results appear robust in terms of their conclusions.

We attach a summary table with all the relevant information of the selected articles (Table 3):

**Table 3 brainsci-12-01570-t003:** Summary table of the 25 studies included in the systematic review.

Year	Author	Title	Main Topic	N	Level	Methodological Approach	Measured Variable	Outcome
1997	Steenland K, Deddens JA [53]	Effect of travel and rest on performance of professional basketball players	Sleep and commuting	8495 NBA season games (1987–1995)	Professional	Each game, one observation (data from the NBA)—regression analysis	Objective: performance statistics	More time between games improves performance. Circadian rhythms positively affect from west to east
2011	Mah et al. [54]	The Effects of Sleep Extension on the Athletic Performance of Collegiate Basketball Players	Sleep and performance	11 male	Collegiate	Fixed-effects linear regression models	Objective: performance on sprints, free throws, 3-pointers, reaction time Subjective: levels of sleepiness and mood	Optimal sleep helps reaching peak performance
2012	Zhao et al. [41]	Red Light and the Sleep Quality and Endurance Performance of Chinese Female Basketball Players	Sleep and performance	20 female	Professional	Cohort study, mixed ANOVA	Subjective: Pittsburgh Sleep Quality Index, Objective: aerobic test	Red light improved quality of sleep
2017	Staunton et al. [55]	Sleep patterns and match performance in elite Australian basketball athletes.	Sleep and performance	17 female	Elite	Prospective cohort study comparing total and quality of sleep vs. performance (bball efficiency statistic)	Objective: triaxial accelerometers and EFF (basketball efficiency)	Game schedule can affect sleep patterns
2017	Tsunoda et al. [44]	Correlation between sleep and psychological mood states in female wheelchair basketball players on a Japanese national team	Sleep and performance	17 female	Elite	Spearman’s correlation + Student’s *t*	Subjective: Pittsburgh Sleep Quality Index (PSQI), Profile of Mood States (POMS-SF)	Vigor is related to sleep and performance
2017	Heishman et al. [42]	Comparing performance during morning vs. Afternoon training sessions in intercollegiate basketball players	Sleep and workloads	10 male	Collegiate	Retrospective study *t*-test	Objective: CMJ and player readiness (Omegawave), Subjective: self-reported sleep quantity	Less sleep, poorer performance
2018	Bonnar et al. [34]	Sleep interventions designed to improve athletic performance and recovery: a systematic review of curret approaches	Sleep and performance	218 athletes, 18–24 yrs, various sports from 10 studies	Collegiate	Systematic review (PRISMA 2016) of PubMed, PsycInfo, and WebofScience	Subjective: sleep interventions survey	Sleep is key for recovery and performance
2018	Thornton et al. [50]	Impact of short- compared to long-haul international travel on the sleep and wellbeing of national wheelchair basketball athletes	Sleep and commuting	11 male	Elite	Linear mixed models determined effects of travel length on sleep and jet lag, fatigue and vigor feelings	Subjective: personal ratings on sleep, jet lag, fatigue and vigor during travel phases, competition phases, and base phases	Long travel is more fatigue-inducing than short travel, affects vigor
2018	Murray et al. [21]	Recovery practices in Division 1 collegiate athletes in North America	Sleep and performance	152 division 1 athletes (bball, football, soccer)	Collegiate	16-item questionnaire	Subjective: attitude towards recovery: sleep, nutrition, cold water immersion, compression garments	Only a few players use sleep as recovery
2018	Mutsuzaki et al. [37]	Comparison of sleep status among three Japanese national wheelchair basketball teams	Sleep and performance	44 Japanese national team wheelchair players: 14 top male, 18 top female and 12U-23 male	Elite	Bonferroni test + Student’s *t*	Subjective: Pittsburgh Sleep Quality Index (PSQI)	Older players sleep worse than young ones. Women have more insomnia than men
2018	Jonathan Roy and Genevieve Forest [44]	Greater circadian disadvantage during evening games for the National Basketball Association (NBA), National Hockey League (NHL) and National Football League (NFL) team travelling westward	Sleep and circadian rhythms	5 years of regular season games (5909 NBA games, 5640 NHL, and 1280 NFL games)	Professional	*T*-tests and analysis of variance and single linear regression models	Objective: teams’ winning percentages	Circadian rhythms affect performance. Disadvantages for teams traveling westward
2019	Jones et al. [56]	Association between late-night tweeting and next-day game performance among professional basketball players	Sleep and performance	112 NBA players	Professional	Merge of 2 public databases of social media and performance	Objective: late night social media activity vs. next day game performance (individual NBA statistics)	Late night social media activity negatively affects performance
2019	Daniel et al. [57]	Effect of the intake of high or low glycemic index high carbohydratemeals on athletes’ sleep quality in pre-game nights	Sleep and performance	9 male	Elite	Cross-sectional study with a crossover design (3-day championship)	Subjective: sleep latency (LAT), sleep efficiency (EFIC), wake after sleep onset (WASO), sleep time through actigraphy and sleep diary, satiety, sleepiness Objective: dietary intake, glycemic response, salivary cortisol and melatonin	Food intake during the day affects night sleep more tan the glycemic index
2019	Clemente et al. [38]	Perceived Training Load, Muscle Soreness, Stress, Fatigue, and Sleep Quality in Professional Basketball: A Full Season Study	Sleep and workloads	15 male	Professional	Descriptive longitudinal study, mixed ANOVA	Subjective: internal load (RPE) and wellness (muscle soreness, stress, fatigue, and sleep quality)	Lower sleep quality and higher fatigue when playing 2 games in a week
2020	Fox et al. [36]	Losing Sleep Over It: Sleep in Basketball Players Affected by Game But Not Training Workloads	Sleep and performance, sleep and workloads	7 male	Semi-pro	Observational study—linear mixed model + Bonferroni tests	Objective: player loads and heart rate Subjective: RPE sleep duration and quality	Higher loads mean shorter sleep duration and lower quality
2020	Fox et al. [48]	The Effect of Game-Related Contextual Factors on Sleep in Basketball Players	Sleep and performance	9 male	Semi-pro	Linear mixed models and effect sizes to compare duration and quality of sleep with game outcomes	Objective: game outcome and score margin vs. subsequent night sleep duration Subjective: sleep quality	Players should sleep more hours after games
2020	Lastella et al. [49]	The Impact of Training Load on Sleep During a 14-Day Training Camp in Elite, Adolescent, Female Basketball Players	Sleep and workloads	11 female	Elite	Separate linear mixed models and effect size analysis assessed differences in sleep behaviors depending on the training load	Objective: wrist activity monitors Subjective: RPE	No effects
2020	Doeven et al. [58]	Managing Load to Optimize Well-Being and Recovery During Short-Term Match Congestion in Elite Basketball.	Sleep and workloads	16 male	Elite	Monitor loads and recovery during a full season	Subjective: RPE and wellbeing (sleep, fatigue, stress, soreness, and mood) during a full season	Workloads negatively affect sleep and wellbeing
2020	Watson et al. [51]	Decreased Sleep Is an Independent Predictor of In-Season Injury in Male Collegiate Basketball Players	Sleep and performance	19 male	Collegiate	Separated mixed-effect logistic regression model	Subjective: mood, fatigue, stress, soreness, Objective: sleep duration (hours), and previous day’s training load and injuries	More sleep, less risk of injury
2020	Cammarano et al. [45]	Sleep and perceived effort during a collegiate women’s basketball season	Sleep and workloads	14 female	Collegiate	Multiple regression analysis	Objective: sleep quantity Subjective: sleep quality (self-reported measures) and RPEs (pre- and in-season)	No effects
2021	Fox et al. [47]	The Association Between Sleep and In-Game Performance in Basketball Players	Sleep and performance	8 male	Semi-pro	Linear regression with cluster-robust standard error to quantify performance–sleep association	Objective: performance (player statistics and composite performance statistics, such as offensive and defensive rating and player efficiency)	Good quality of sleep before a game, better performance
2021	Filipas et al. [52]	Single and Combined Effect of Acute Sleep Restriction and Mental Fatigue on Basketball Free-Throw Performance	Sleep and performance	19 male	Amateur	2 identical experimental sessions, 1 control vs. 1 with MF and SR	Subjective: mental fatigue Objective: % free throws (SR and MF alone and combined)	Better sleep, better % free throws
2021	Conte et al. [46]	Workload and well-being across games played on consecutive days during in-season phase in basketball players	Sleep and workloads	7 male	Semi-pro	Linear mixed model with fixed effect	Objective: PL (microsensors), heart rate, HR zones Subjective: RPE, wellbeing: fatigue, sleep quality, soreness, stress, and mood (survey)	Lower sleep, recovery, and wellbeing, higher fatigue after 2 consecutive games
2021	Raya et al. [59]	Caffeine Ingestion Improves Performance During Fitness Tests but Does Not Alter Activity During Simulated Games in Professional Basketball Players	Sleep and performance	14 male	Professional	Double-blind, counterbalanced, randomized, crossover study	Objective: physical tests (CMJ, agility drills, sprint tests)	Caffeine does not improve basketball peformance but promotes sleep perturbance
2021	Charest et al. [60]	Impacts of travel distance and travel direction on back-to-back games in the National Basketball Association	Sleep and circadian rhythms, sleep and performance	NBA	Professional	8 years of NBA seasons (2013–2020)	Objective: number of flights, the timing of flights, timings of arrival at destination and hotel, sleep time	Travel fatigue and circadian desynchronization disturb sleep and recovery

NBA: National Basketball Association, NHL: National Hockey League, NFL: National Football Association, RPE: rating of perceived exertion, EFF: Basketball Efficiency statistic, CMJ: countermovement jump, MF: mental fatigue, SR: sleep restriction, PL: player load, HR: heart rate, LAT: sleep latency, EFIC: sleep efficiency, WASO: wake after sleep onset.

## 4. Results

The initial search of scientific literature related to our systematic review presented a result of 50 total articles, of which a total of 25 were excluded from the study, as they were not relevant or related to sleep and its role in basketball (they did not meet the inclusion criteria) (Figure 1).

### 4.1. Main Search

The variables analyzed in these studies focused on the comparison of different sleep interventions (such as sleep quality and extension, sleep hygiene, the use of electronic devices during the night…) with the performance of athletes, measured with different magnitudes of basketball, all of them typical of this sport. This type of relationship has been found in 8 articles [35,38,45,48,49,54,56,58]. We have also seen an important emphasis in the study of the relationship of workloads and the quality and extent of sleep of athletes (6 articles were dedicated to this topic [37,39,46,47,50,57]), as well as in the relationship of travel, jet-lag and traveler fatigue and its effect on the sleep of the athlete (9 articles analyzed these topics). Also related to circadian rhythms and their association with basketball performance, a study analyzed the best times of the day to train based on these rhythms, comparing the consequences of training in the morning with evening workouts [43]. On the other hand, one study has linked sleep and workloads to the possibility of injury of the athlete [52] and two works have related nutrition and ergogenic aids with the subsequent quality and extension of sleep [53,59]. 

### 4.2. Assessment of the Quality of the Studies

As for the participants of the studies, we divide them among empirical studies with athletes of different characteristics (professionals vs. amateur) and reviews of studies or complete seasons of a specific professional competition. All these studies provide us with relevant information, so they have been included in our review. Regarding the professional competitions analyzed, 4 reviewed articles contain 99 studies on the subject and a total of 14,404 NBA games analyzed [18,44,54,60]. Furthermore, the newest study from Charest et al [61] analyzed 8 full NBA seasons. From the empirical studies carried out on different athletes, we have divided the participants into basketball players and athletes (team sports but not necessarily basketball). The studies conducted directly on basketball players include a total of 69 amateur players (17 women and 52 men), 54 NCAA Division I players (14 women and 40 men), 72 elite wheelchair basketball players (35 women and 37 men), 105 elite and semi-professional players (40 women and 65 men) and 161 professional players (20 women and 141 men, of which 112 are NBA players). As for the studies of athletes (usually of team sports), these gathered 745 participants, of which 266 were women and 479 men).

Therefore, there is a clear bias in the studies towards the inclusion of a majority of male athletes. We located one study [40] that precisely details the difference in sleep between male and female athletes. However, this study was discarded because of the use of the REST-Q scale as a sleep measure [55].

As for the chronology of the publications studied (Figure 2), it should be noted that of these 22 most recent articles, 15 have been published in the last two years (8 in 2020 [37,41,46,48,49,50,52,57] and 7 in 2021 [18,34,36,47,53,56,61]). The rest of the articles correspond to 6 studies of 2018 [22,26,34,38,44,51] and 3 of 2019 [47,54,59]. Besides, we have examined 3 articles of 2017 [43,45,62] and of previous years, we have examined an article of 1997 [60], another of 2011 [58] and another of 2012 [42]. We see therefore that researchers in sports sciences are taking the study of sleep very seriously, which proves to be one of the keys in the health and performance of athletes [35,63].

### 4.3. Risk of Bias

Risk of bias are the likelihood that features of any study may give misleading results. As such, we analyzed the following: in relation to the risk of selection bias (Table 4), 18 studies have been categorized as low risk [18,26,38,39,42,43,44,47,51,52,53,54,56,57,58,60,61,62], 8 as medium risk [22,35,37,45,48,49,50,59] and two high-risk [18,46]. Regarding the risk of realization bias, one study was classified as high risk [18], 6 studies as medium risk [37,45,46,48,49,59], while all other studies were categorized as low risk. One study has been categorized as high risk of detection bias [18], 8 medium risks [37,45,46,48,49,50,54,59], with the rest of the studies classified as low risk. If we talk about the risk of assertion bias, two studies [18,46] have been classified as high risk, 6 intermediate risk studies [37,45,46,49,54,59] and the rest have been considered as low risk. In addition, in reference to reporting risk bias, two studies are at high risk [18,46] and 4 medium risk [50,54,59,60] and the rest are considered to be low risk. And finally, in reference to other risks of bias, only two studies [18,46] have been considered high risk and 7 others medium risk [35,38,42,43,51,54,59] (Figure 3).

All the information analyzed from these 25 articles reveals that the conclusions reached by the different authors after their respective analyses are significantly homogeneous. We could say that, in general, studies present a clear relationship between sleep and sports performance, in this case in basketball. There are different topics addressed in relation to sleep and performance, which we could classify as follows:First, 8 studies concluded that the relationship between performance and the quantity and quality of sleep is directly proportional, and that, therefore, the greater the hours of sleep and the greater the continuity of sleep across a sleep period, the subsequent performance on the basketball court increases [22,35,36,48,49,54,56,58].

Two of them also reveal that among young (university) players, men recover better than women [40] and, in addition, that although young players are aware that sleep is key in recovery, only 24% of them consider it as such [22].

Besides, 6 studies focused on training or match loads applied to players, being four of them related to high stress loads with worse nights’ sleep [37,39,47,57], while only one study did not find this relationship [50]. Another study [46] found no relationship between sleep quality and associated Rate of Perceived Exertion (RPE) of subsequent training (that is, it is concluded that high training loads affect the quality of sleep on the night following the training session, so performance the next day will be negatively affected). This should be taken into account when planning recovery for these tougher sessions, where it would be convenient to lengthen the number of hours slept by players.Only one study examined the likelihood of injury compared to the number of hours slept, concluding that more sleep equates to reducing future chances of injury [52].Six articles studied the relationship between travel and circadian rhythms, concluding that traveling, due to fatigue, jet lag, time changes etc. significantly negatively affect the sleep, recovery and performance of players [18,26,44,51,60,61]. One of the studies highlights that the relationship of performance and sleep is affected differently to each individual [62], so each coaching staff should know this individual variability for their players. On the other hand, another study [43] highlights how circadian rhythms affect daily energy peaks, underpinning the fact that you train and compete better in the afternoon than in the morning. This information is also closely related to the calendars of professional teams, being the NBA and their trips to different time zones its paradigmatic case. This should be taken into account in the future to improve the quality of life of the players themselves, as well as their performance and their risk of injury.

In 3 of these articles, the importance of the competitive calendar was also highlighted, in a similar way, and how it negatively affects sleep, recovery and performance [18,26,51] due to its corresponding saturation of matches and the short time to recover between them, especially when playing "back-to-back" matches.

3 articles linked sleep quality to vigor and mood in athletes with disabilities (wheelchair basketball players) [38,45] or isolated by Covid19 [36].In two articles that related sleep to nutrition and ergogenic aids, it was concluded that caffeine does not affect basketball performance [53] and that the food that the athlete eats for an entire day affects sleep more than the glycemic index of his last meal, dinner [59].One article concludes that red light therapies before sleep affect athletes in a positive manner, allowing them a better quality of sleep and a better recovery after training or match [42]. This type of therapy could be very useful to improve the recovery of athletes in very saturated periods of competition or when the competitive calendar forces them to travel continuously.

After having analyzed the 25 publications mentioned, we found that despite the existing attempts to modernize and improve basketball analysis [64,65], scientific information about sleep and its practices does not seem to have yet reached the offices of clubs or even professional basketball competition organizations. We believe that the practical applications derived from the conclusions of this review should be taken into account by clubs and leagues, and that this could have very positive consequences on the health of the players and their performance, and therefore, on quality and spectacle and even on long-term economic viability.

All the above information is summarized in the following Table 5:

## 5. Discussion

The present systematic review has focused on reviewing the scientific work that has been published regarding the effect that sleep can have on the performance of basketball players. After analyzing 25 articles, the majority of them published in recent years, we can conclude that sleep does affect basketball performance and its related injuries. In addition, the results show that there are several factors, from trips to external and internal loads or training schedules, that negatively affect the rest of the athlete and that are not being considered when applying recovery strategies.

### 5.1. Sleep and Performance

The main conclusion of this systematic review is that sleep does have a clear impact on performance in the sport of basketball [34], affecting not only the statistics of the players [48,54,56,58] but also to the chances of injury [52]. Therefore, it seems undeniable that sleep should be a fundamental control variable by the coaching staff of any team [66].

### 5.2. Sleep and Workloads

The results of the studies analyzed show that the loads of the workouts affect sleep in the following way: higher loads prevent athletes from having quality sleep that same night [36]. This affects performance the next day [39,47,57], so coaches should take this into account when planning and deciding their workout intensity and schedules. 

### 5.3. Sleep and Circadian Rhythm

Circadian rhythms affect all players and coaches and they determine the best and worst times of the day to train and compete [43]. This is not being taken into account when scheduling training or competitions [62]. If this variable were taken into account consistently, the performance of players could be increased in a similar way. Future lines of research should focus on this to further develop this idea.

### 5.4. Sleep and Traveling

In professional competitions, it is increasingly common to travel hundreds or thousands of kilometers per week in order to meet the assigned matches [26,61]. This has a significant toll players health and performance: long journeys negatively affect recovery of athletes, through factors such as the physical toil of travel and disruptions to the circadian rhythm [18,51,60]. Therefore, the performance of the players tends to be worse and the probability of injuries, higher. Again, this is not being taken into account when choosing competition calendars. The respective organizations should take this into account to improve competitions and better protect players, who are the most important assets that clubs have [18].

## 6. Conclusions

This work of review of the most current scientific literature shows that sleep and circadian rhythms are very fashionable topics that are acquiring a lot of importance in the competitive environment of basketball in recent seasons. There are finally solid scientific conclusions that crown sleep as a basic practice of recovery of the athlete and it is clear that it affects their performance and health. Dismissing the use of good sleep practices and not having into account the diverse circadian rhythms of our athletes can have deleterious consequences on basketball performance.

## 7. Practical Applications

Education in sleep hygiene and its important consequences should come from the hand of the staff of each team, to enhance the only natural (and free) tool that exists to recover the body from the stress to which we subject it in each training and game [67,68]. Players must understand how to create a positive pre-sleep environment (avoiding the use of stimulants, social media and bright light before going to bed, especially blue light, using amber tinted glasses and night-shift-mode for the electric devices as a countermeasure, setting a cooler and quiet environment without distractions) and how this can affect their performance, both in numbers and statistics and in injuries.

On the other hand, coaches should take more into account natural circadian rhythms when scheduling loads for both morning and evening workouts, allowing more hours of sleep for their players after a particularly hard training day or match. The same applies after a long and heavy trip, when facing the next training. Sleep should be considered as one more recovery practice and therefore should have the same importance as the usual practices in these cases (cold immersion, ergogenic aids, massages, compression garments…).

As for professional basketball clubs and leagues, it seems necessary that more attention be paid to the negative impact of travel on the performance and health of players. Now that it is clearer how they suffer in their performance and how the chances of injury increase, both factors that condition the spectacle of this sport and thus, are key when it comes to attracting fans (and therefore, the money necessary to keep everything moving), sleep cannot be considered trivial. Moreover, in the main professional leagues of the world (NBA, Euro league), where long trips to different time zones are common, emphasis should be placed on providing enough hours of rest between trips, on avoiding the classic schedule of two consecutive games (back-to-back games) and even the time of the game itself. We must remember that in the NBA it has been shown [44] how when a team from the West travels to the East the time of the game benefits him by coinciding with its peak of the circadian rhythm. We believe that all these recommendations can increase the value of the professional sport of basketball at several levels, from the purely sports’ side to the organizational and even throughout the economic part. By improving both the show and the organization of it, it must necessarily result in a greater attraction on the part of the spectator and a more efficient use of budgets by the clubs.

All the information collected in this work should be taken into account by both the players and the coaches of the teams (staff members are also affected by these conditions, as Calleja-González et al pointed out in 2021 [66]), the managers of the clubs and by the organizations of the professional leagues themselves, in order to facilitate a safer and more coherent environment with the circadian rhythms and recovery times of the human being.

## 8. Future Lines of Research

Basketball is a sport that requires more accurate future research on sleep and with a larger population and duration over time. We know that at the elite of the sport, the NBA [26], attention is beginning to be drawn to such an important issue [18] and we believe that progress has to be made in that direction. The approach to sleep, therefore, should be personalized, always taking into account the individuality of each player and being able to provide a tailor-made solution. 

## 9. Limitations

The number of articles related to this topic that have been based on professional players is very small, due to the difficulty of examining this type of profile. That is why in this review it has been decided to also include the information provided by the studies with semi-professional and amateur players.

Regarding the risk of bias and the quality of the methodology used, disagreements have been discussed between the two authors, JOL and JC-G, until reaching an agreement on the subject.

## Figures and Tables

**Figure 1 brainsci-12-01570-f001:**
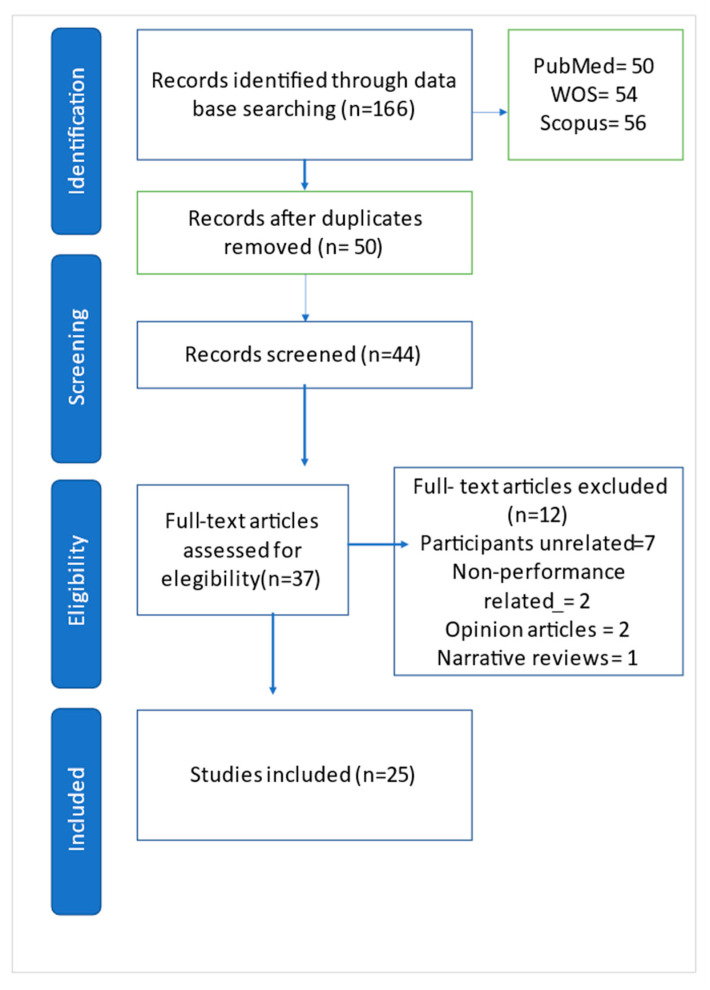
Selection of studies (PRISMA, 2009 Flow Diagram).

**Figure 2 brainsci-12-01570-f002:**
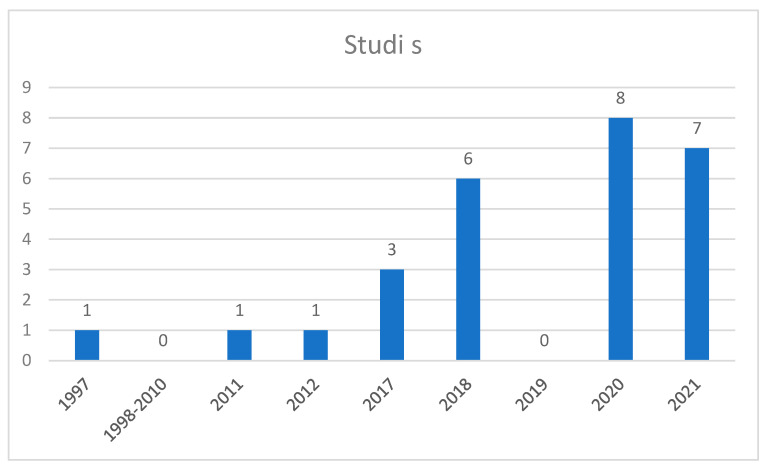
Chronology of the number of sleep-related published studies.

**Figure 3 brainsci-12-01570-f003:**
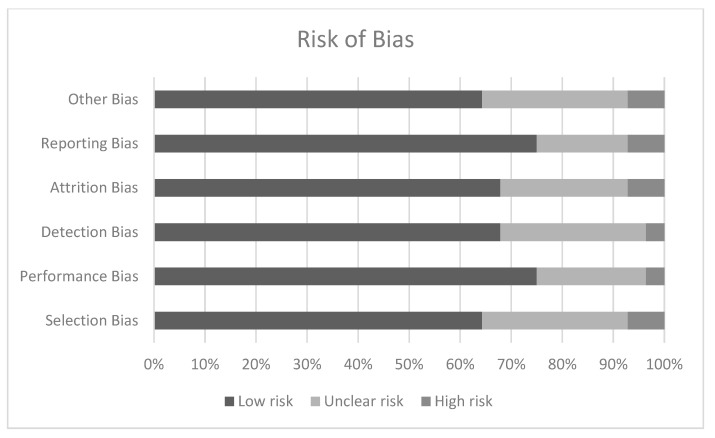
Graph of Risk of Bias.

**Table 1 brainsci-12-01570-t001:** OLE Scale.

Level	Evidence
*Level 1*	Meta-analysis of high-quality randomized controlled trials (RCTs) or RCTs
*Level 2*	Lower quality RCTs or prospective comparative studies
*Level 3*	Case studies or retrospective studies
*Level 4*	Cases without comparison of control groups
*Level 5*	Case reports or expert opinions

**Table 2 brainsci-12-01570-t002:** Applied OLE Scale.

OLE	Nº Studies
L1	3
L2	0
L3	4
L4	14
L5	4

**Table 4 brainsci-12-01570-t004:** Summary showing Risk of Bias.

Year	Author	Selection Bias	Performance Bias	Detection Bias	Attrition Bias	Reporting Bias	Other Bias
1997	Steenland et al [59]	+	+	+	+	?	+
2011	Mah et al [55]	+	+	+	+	+	+
2012	Zhao at al [42]	+	+	+	+	+	?
2017	Staunton et al [62]	+	+	+	+	+	+
2017	Tsunoda et al [45]	?	?	?	?	+	+
2017	Heishman et al [43]	+	+	+	+	+	?
2018	Bonnar et al [35]	?	+	+	+	+	?
2018	Thornton et al [51]	+	+	+	+	+	?
2018	Murray et al [22]	?	+	+	+	+	+
2018	Mutsuzaki et al [38]	+	+	+	+	+	?
2018	Roy J and Forest G [44]	+	+	+	+	+	+
2018	Huygue et al [26]	+	+	+	+	+	?
2019	Jones et al [54]	+	+	?	?	?	?
2019	Daniel et al [57]	?	?	?	?	?	?
2019	Clemente et al [39]	+	+	+	+	+	+
2020	Fox et al [37]	?	?	?	?	+	+
2020	Fox et al [49]	?	?	?	?	+	+
2020	Lastella et al [50]	?	+	?	+	?	+
2020	Doeven et al [56]	+	+	+	+	+	+
2020	Watson et al [52]	+	+	+	+	+	+
2020	Cammarano et al [46]	-	?	?	-	-	-
2021	Fox et al [48]	?	?	?	?	+	+
2021	Filipas et al [53]	+	+	+	+	+	+
2021	Singh et al [18]	+	+	+	+	+	+
2021	Conte et al [47]	+	+	+	+	+	+
2021	Raya-González et al [58]	+	+	+	+	+	+
2021	Singh et al [34]	-	-	-	-	-	-
2021	Charest et al [60]	+	+	+	+	+	+

Effects of sleep on performance.

**Table 5 brainsci-12-01570-t005:** Summary of sleep-related fields.

Nº Studies	Thematic	Conclusion
9	Sleep and performance	More and better sleep, better performance
7	Sleep and workloads	High loads, worse subsequent sleep and worse recovery
6	Sleep and circadian rhythms	Circadian rhythm affects performance
3	Sleep and traveling	Travel increases tiredness and worsens recovery

## Data Availability

The data presented in this study are available in the article.

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
