# Peer review of "How Sleep Affects Recovery and Performance in Basketball: A Systematic Review"

_brainsci, 2022, doi:10.3390/brainsci12111570_

Round 1

Reviewer 1 Report

Manuscript ID: brainsci-2020227

Title: "How sleep affects recovery and performance in basketball. A systematic review"

The aim of this review was to evaluate the effects of sleep on basketball players.

To this end, 25 studies were selected. Overall, results confirm an important role of sleep in basketball performance.

The topic of the present study is interesting, the methodology clear and compliant with international standards.

I only have few suggestions.

In Table 3, the kind of instruments (or variable) adopted is not reported. It could be useful to easily distinguish between subjective and objective measures. In the present form, it is very hard to understand this feature.

Table 4 could be deleted, in alternative I suggest inserting a space between 1997 and 2011, to underline the time elapsed.

In Table 5 I suggest changing the colours, in order to allow a clear view even in a black and white print.

In the conclusions section, Authors could get back to the theme linked to the nature (objective and / or subjective) of the evaluation of sleep and circadian rhythms.

Author Response

Dear reviewer,

Thank you for your report. We agree with al your comments and, as such, we changed the text to include your suggestions (all changes in red)

Point 1: Table 3 - we agree with the suggestion, so we included a distintion between Subjective and Objective measures.

Point 2: Table 4 - we agree with the suggestion, so we included a gap for the period 1998-2010, so it's easier for the reader to see that we did not find relevant sleep science from those years

Point 3: Table 5 - We agree with the suggestion and we changed the colours of the bars so they can allow a clear view.

Point 4: Conclusion - We agree with the suggestion, so we added more references to our main topic (sleep and circadian rythms).

Again, thank you for the proposed changes.

Reviewer 2 Report

The manuscript is a comprehensive systematic review focused on the effect of sleep on the performance of basketball players. The authors summarize recent literature on the effect of sleep on the performance of basketball players and providing the recommendations to develop intervention strategies to improve sleep habits among the athletes. The review highlights the importance of sleep health for basketball players and will contribute the sports community awareness on this issue.

This article gives general background to understand role of sleep on the athletes performance. The review is quite interesting and very well written, the bibliography is extensive and up to date.

However, I would suggest the authors to correct several typos in the manuscript some of them I am addressing below:

1.First of all I have to mention that the manuscript is missing page numbers and lines from the beginning.

2.      page 2 “sleep practices of athletes” for me sounds better “sleep habits of athletes”

3.      page 2 "the probabilities of a solid and systematic sleep are scarce for elite basketball players " Please, explain what do you mean under the solid and systematic sleep? I think its better to  reformulate.

4.      pg 3, line  6    please, instead of “sleepiness” use “daytime sleepiness”

5.      pg 7 table 3 needs adjustment, in the column of the methods you report just statistics used in the work. It would be more beneficial for readers to have the information about the methodological approaches in each article.

6.      pg15 line- 46 missing year

7.      pg16 line 61 "1524" missing comma

8.      pg 17 line 104 missing punctuation – dot

9.      Pg18 line 118 "1article" please, correct "One article"

10.   pg 18 line 150 "deciding their workout schedules"  please, replace with  "deciding their workout intensity and schedules"

11.   pg.19 line 179 before the bedtime to avoid the bright light especially blue light as a countermeasure can be used amber tinted glasses and night shift mode for the electric devices.

12.   pg 19 line-207 reference 8 should be in the brackets.

13.   The references are not presented according the MDPI style.

Author Response

Dear reviewer,

Thank you for your report. We agree with al your comments and, as such, we changed the text to include your suggestions (all changes in red)

Point 1: we agree with the suggestion, and include the proposed change (numbers and lines)

Point 2: we agree with the suggestion, and include the proposed change (habits included)

Point 3: we agree with the suggestion, and include the proposed change (reformulation)

Point 4: we agree with the suggestion, and include the proposed change (daytime included)

Point 5: we agree with the suggestion, and include the proposed change. Column name changed, also including objective and subjective measured variables.

Point 6. we agree with the suggestion, and include the proposed change

Point 7: we agree with the suggestion, and include the proposed change (comma)

Point 8: we agree with the suggestion, and include the proposed change (dot)

Point 9: we agree with the suggestion, and include the proposed change

Point 10: we agree with the suggestion, and include the proposed change

Point 11: we agree with the suggestion, and include the proposed change

Point 12: we agree with the suggestion, and include the proposed change (brackets)

Point 13: we agree with the suggestion, and include the proposed change, presenting the references according to the MDPI style

Again, thank you for the proposed changes.

Kind regards,